# Structures of human SGLT in the occluded state reveal conformational changes during sugar transport

Wenhao Cui [1,2,5], Yange Niu [1,2,5], Zejian Sun[3,4], Rui Liu [1,2] & Lei Chen [1,2,4]

Sodium-Glucose Cotransporters (SGLT) mediate the uphill uptake of extracellular sugars and play fundamental roles in sugar metabolism. Although their structures in inward-open and outward-open conformations are emerging from structural studies, the trajectory of how SGLTs transit from the outward-facing to the inward-facing conformation remains unknown. Here, we present the cryo-EM structures of human SGLT1 and SGLT2 in the substrate-bound state. Both structures show an occluded conformation, with not only the extracellular gate but also the intracellular gate tightly sealed. The sugar substrate are caged inside a cavity surrounded by TM1, TM2, TM3, TM6, TM7, and TM10. Further structural analysis reveals the conformational changes associated with the binding and release of substrates. These structures fill a gap in our understanding of the structural mechanisms of SGLT transporters.

Sugars, such as glucose, are one of the most important nutrients across all kingdoms of life. The uptake of glucose against its concentration gradient is mediated by the sodium-dependent glucose cotransporters (SGLT)[1]. These transporters utilize the electrochemical gradient to drive the uphill transport of glucose and play essential roles in human health[1,2]. SGLT1 (SLC5A1) and SGLT2 (SLC5A2) are two important subtypes of this family, and their physiological functions are well-documented. The loss-of-function mutations of genes encoding SGLT1 and SGLT2 could lead to intestinal glucose-galactose malabsorption and familial renal glucosuria in humans, respectively[1]. SGLT2-specific inhibitors are widely prescribed for the treatment of diabetes[3]. Moreover, SGLT inhibitors also show promise in the treatment of cardiovascular diseases, constipation, and certain types of cancer[4–7]. Pioneering structural work on the bacterial homolog vSGLT from *Vibrio parahaemolyticus* revealed that SGLT proteins have 14 transmembrane helices, which assemble into a LeuT-like fold[8]. In the presence of substrate and sodium, vSGLT resembles an inward-facing conformation[8]. The recent development of cryo-EM facilitates the structure determination of human SGLT (hSGLT). A consensus-mutated hSGLT1 in the apo state shows an inward-open structure (hSGLT1$_{inward-open}$)[9]. The SGLT2-specific inhibitor empagliflozin and

the non-specific inhibitor LX2761 occupy not only the sugar substrate-binding site but also the extracellular vestibule to lock hSGLT2 and hSGLT1 in the outward-open conformation[10,11]. Despite this progress, the structure of SGLT in the occluded conformation is yet unknown, representing a key missing state during the transporting cycle of SGLT, which precludes a comprehensive mechanistic understanding of how SGLT functions.

In this work, we present the cryo-EM structures of hSGLT1 and hSGLT2 in the substrate-bound occluded conformation. We also show the structure of hSGLT2 in the apo state in an inward-open conformation. Structural comparisons of SGLT between states reveal how the essential structural elements concertedly move during the transport cycle.

## Results

### Structures of hSGLT1 and hSGLT2 in the substrate-bound state

We adopted the "three-joint-tethering" strategy to generate the fully functional hSGLT1$_{GFP}$-MAP17$_{nb}$ protein for structural studies[10–12]. It was reported that replacing the equatorial hydroxyl group at the 4-position of glucose with a fluorine atom yielded a glucose analogue, 4-deoxy-4-fluoro-D-glucose (4D4FDG), with an enhanced affinity for SGLT1

[1]State Key Laboratory of Membrane Biology, College of Future Technology, Institute of Molecular Medicine, Peking University, Beijing Key Laboratory of Cardiometabolic Molecular Medicine, 100871 Beijing, China. [2]National Biomedical Imaging Center, Peking University, 100871 Beijing, China. [3]Academy for Advanced Interdisciplinary Studies, Peking University, 100871 Beijing, China. [4]Peking-Tsinghua Center for Life Sciences, Peking University, 100871 Beijing, China. [5]These authors contributed equally: Wenhao Cui, Yange Niu. ✉e-mail: chenlei2016@pku.edu.cn

($K_{0.5}$ is 0.07 mM) compared with glucose ($K_{0.5}$ is 0.5 mM) (Supplementary Fig. 1a, b)[13]. We found that 4D4FDG inhibits the 1-NBD-glucose uptake of the hSGLT1$_{GFP}$-MAP17$_{nb}$ construct with an IC$_{50}$ around 0.07 mM (Fig. 1a). Therefore, to obtain the structure of hSGLT1 in the substrate-bound state, we supplemented the hSGLT1$_{GFP}$-MAP17$_{nb}$ protein with 4D4FDG throughout the purification, the nanodisc reconstitution, and the cryo-EM sample preparation (Supplementary Fig. 1). Single particle analysis yielded a reconstruction of the hSGLT1-4D4FDG complex at 3.26 Å resolution (Supplementary Figs. 2 and 3 and Supplementary Table 1). A ligand density was found at the sugar-binding pocket of hSGLT1 (Fig. 1b–d). This density was not observed in the structure of hSGLT1 in the absence of substrate[9], confirming the identity as the 4D4FDG molecule. We modeled the 4D4FDG into the density according to the structures of the vSGLT-galactose complex[8] and the hSGLT1-LX2761 complex[11] (Supplementary Fig. 3b, c). To visualize the substrate-bound structure of hSGLT2, we supplemented methyl alpha-D-galactopyranoside (AMG) into the hSGLT2$_{GFP}$-MAP17$_{nb}$ protein in nanodisc (Supplementary Fig. 1f, g). Subsequent single particle analysis generated reconstruction of the hSGLT2-AMG complex at 3.33 Å resolution (Supplementary Figs. 4 and 5 and Supplementary Table 1). We observed that AMG is bound in an occluded substrate-binding site (Supplementary Fig. 5a) and we modeled the AMG into the density according to the structures of the vSGLT-galactose complex[8] and the hSGLT2-empagliflozin complex[10] (Supplementary Fig. 5b, c). The structure and conformation of the hSGLT2-

AMG complex are overall similar to those of the hSGLT1-4D4FDG complex (Supplementary Fig. 5d). We did not observe densities for sodium ions in these structures, likely due to their low resolutions. Notably, although the structural models, especially the coordinates of sugar substrates, were refined against the cryo-EM maps to reasonable geometry, we suggest a cautious interpretation of the sugar-binding pose, because of the small size of sugars, the indistinguishable density of their hydroxyl groups, and the large positional uncertainty intrinsic to this resolution range (~3.3 Å). Therefore, we focus our discussion on the conformational changes of SGLTs instead of the detailed sugar substrates coordination. We mainly use the structure of the hSGLT1-4D4FDG complex for analysis because of its slightly higher resolution (Supplementary Table 1), unless indicated otherwise.

## hSGLT1 in the occluded conformation

The surface representation of the hSGLT1-4D4FDG complex shows the substrate-binding site is occluded to both the extracellular and intracellular solvent (Fig. 2a). 4D4FDG is caged inside a cavity surrounded by TM1, TM2, TM3, TM6, TM7, and TM10 (Fig. 2b). Above the sugar-binding site, I98 and F101 on TM2, M283 on TM6, and F453 and Q457 on TM10 block the substrate entry pathway from the extracellular solution (Fig. 2c). Below the sugar-binding site, N78 on TM1, Y290 and W291 on TM6 form the first layer of insulation. S77 on TM1, V296 on TM6, and S396 on TM8 form the second layer of insulation. These two layers block the substrate release pathway to the cytosol (Fig. 2d).

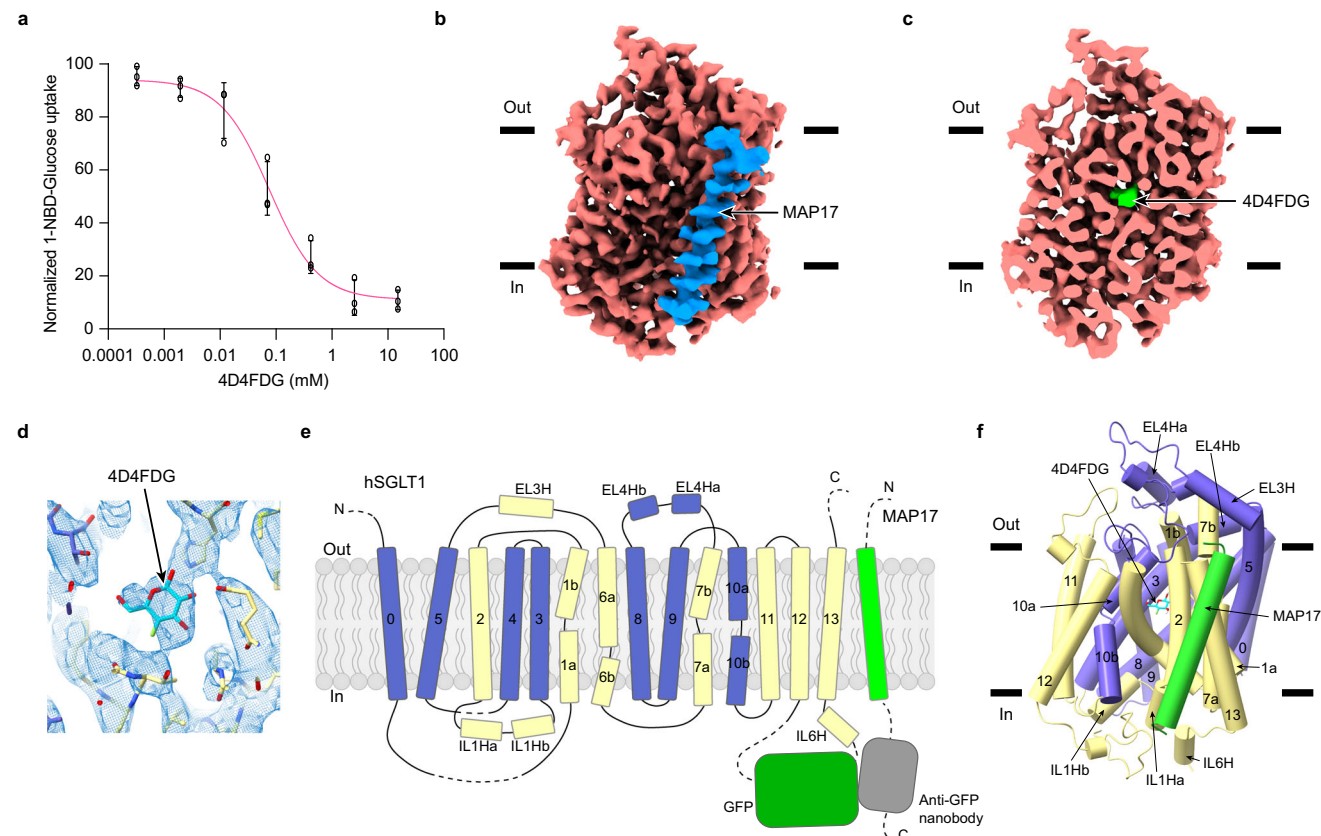

**Fig. 1 | Cryo-EM structure of human SGLT1-MAP17 complex. a** The inhibition curve of the hSGLT1-MAP17 complex by 4D4FDG in the 1-NBD-glucose uptake assay (data are shown as means ± standard deviations; *n* = 3 biologically independent experiments). The 1-NBD-glucose uptake was normalized to the uptake of hSGLT1$_{GFP}$-MAP17$_{nb}$. Source data are provided as a Source Data file. **b** Cryo-EM density map of the hSGLT1$_{GFP}$-MAP17$_{nb}$ complex. hSGLT1 is colored in red. MAP17 is colored in blue. The density of the detergent micelles has been omitted for clarity. **c** The cut-open view of the hSGLT1$_{GFP}$-MAP17$_{nb}$ complex shows the binding site of

4D4FDG substrate inside hSGLT1. 4D4FDG is colored in green. The hSGLT1 is colored the same as in (**b**). **d** The electron density of 4D4FDG and nearby residues. **e** Topology of the hSGLT1-MAP17 complex. The unsolved regions are shown as dashed lines. The moving region of hSGLT1 is colored in blue and the less-mobile region is colored in yellow. **f** The hSGLT1-MAP17 complex in cartoon representation, colored the same as in (**e**). Helices are shown as cylinders. 4D4FDG is shown as sticks.

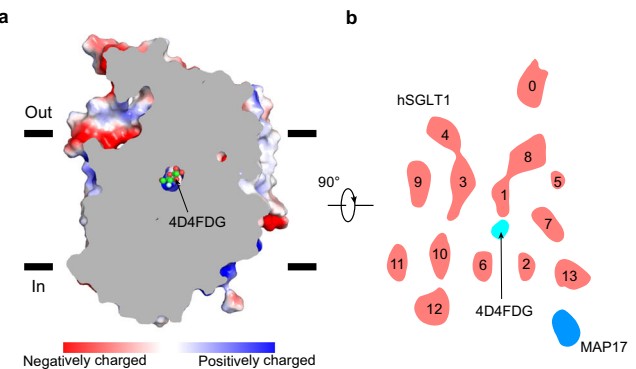
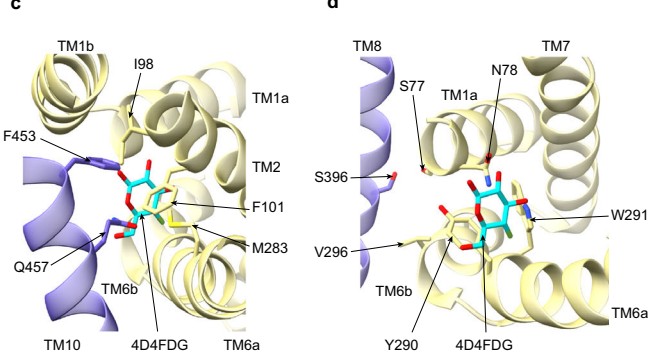

**Fig. 2 | The closed intracellular gate and extracellular gate in the occluded state. a** The estimated surface potential of the hSGLT1-MAP17 complex. 4D4FDG is shown as spheres. **b** Top view of the cross-section of the transmembrane domain of the hSGLT1-MAP17 complex cryo-EM map, colored the same as in Fig. 1b. The map

was low-pass filtered at 7 Å for representation. The numbers of transmembrane helices are labeled above each helix. **c** Residues forming the extracellular gate. **d** Residues forming the intracellular gate. 4D4FDG and gating residues are shown as sticks.

## Structural changes from the outward-open conformation to the occluded conformation

To reveal the conformational change of hSGLT1 from the outward-open conformation to the occluded conformation, we compared the outward-open structure of hSGLT1 with the occluded structure (the hSGLT1-4D4FDG complex). Since the structure of hSGLT in the outward-open state without any ligand bound is not available, we use the LX2761-bound hSGLT1 structure[11] as the reference for the outward-open state because the inhibitor LX2761 opens its sugar entry pathway. Similar to our previous observation in the structural comparison of hSGLT1 between the outward-open and inward-open states[11], the structures of TM0, TM3, TM4, TM5, TM8, TM9, and TM10 have marked movements and are therefore designated as the "moving region", while the movements of TM1, TM2, TM6, TM7, TM11, TM12, and TM13 are less pronounced, and we designated them as the "less-mobile region" (Supplementary Fig. 6). Notably, the moving region contains the "hash motif" (TM3, TM4, TM8, and TM9), and the less-mobile region contains the "bundle domain" (TM1, TM2, TM6, and TM7) and the "gating helix" (TM5 and TM10) proposed for vSGLT[14]. In the following structural comparisons, we aligned the less-mobile region between states to reveal the marked structural changes of the moving region, unless indicated otherwise. We found that the helices of the moving region move inwardly to close the extracellular gate of hSGLT1 upon substrate-binding (Fig. 3a–d). In the extracellular vestibule, F453 moves 3.8 Å inwardly due to the rotation of TM10a and forms hydrophobic interactions with I98 and F101 on TM2 to close the extracellular gate (Fig. 3e). In addition, the 17° inward rotation of TM10a, which is involved in substrate binding, is associated with the counterclockwise rotation of TM4 (13°) and TM9 (15°), viewed from the extracellular side (Fig. 3b). These conformational changes result in a tighter packing of the Na2 site (Fig. 3f) and sugar substrate-binding site, as indicated by the shortened distance between adjacent structural elements (Fig. 3g), while the Na3 site is almost unchanged (Fig. 3h).

## Structural changes from the occluded conformation to the inward-open conformation

To uncover the conformation of hSGLT2 in the apo state, we determined its cryo-EM structure to 3.48 Å resolution (Supplementary Fig. 7 and Supplementary Table 1), which aligns well with the apo hSGLT1 structure in the inward-facing conformation[9] (Supplementary Fig. 8b), suggesting that the hSGLT2 in the apo state also shows an inward-facing conformation, the same as hSGLT1. To reveal how SGLT transits from the occluded conformation to the inward-facing conformation to release the sugar substrates into the cytosol, we compared the structure of the hSGLT1-4D4FDG complex (occluded) with that of the apo

hSGLT1 (inward-open conformation)[9] (Fig. 4). Again, we observed little changes in the less-mobile region but large movements in the moving region (Supplementary Fig. 6). Both TM5 and TM8 move outwardly, together with the associated movements of TM3 and TM10 (Fig. 4a–d). As a result, the local structures of the Na2 and Na3 sodium-binding sites are disrupted (Fig. 4e–f), and the central substrate-binding cavity is enlarged (Fig. 4g). These changes would likely result in the decreased affinity of sodium ions and substrates for hSGLT1. Moreover, TM8 moves outward, resulting in a large outward shift of S396 (Fig. 4h), which participates in the insulation of the sugar-binding pocket from the cytosol in the occluded state (Fig. 2d). As a result, the intracellular gate is partially opened, allowing the release of sugar and sodium ions into the cytosol.

## MAP17 facilitates the surface expression of hSGLT2

MAP17 is an essential subunit for the glucose uptake activity of hSGLT2 but not hSGLT1[15]. However, how MAP17 activates hSGLT2 remains unknown. It is possible that MAP17 activates hSGLT2 by facilitating its conformational changes during the transport cycle or by enhancing its surface expression. Because MAP17 interacts with TM13 of hSGLT2, which is part of the less-mobile region that shows little conformational changes between different states of the hSGLT2-MAP17 complex, the conformation of MAP17-TM13 also stays the same (Fig. 5a, b). Similar observations can be found in the hSGLT1-MAP17 structure (Fig. 5c), suggesting MAP17 might play a structural role in the hSGLT-MAP17 complex. To study whether MAP17 could facilitate the surface expression of hSGLT2, we generated a monoclonal antibody, mAb90, which recognizes an extracellular epitope of hSGLT2 (Fig. 5d). Labeling of hSGLT2 expressed in intact cells using mAb90 showed that the surface expression of hSGLT2 was greatly enhanced by co-expression of MAP17 (Fig. 5d), in agreement with the markedly increased uptake activity of hSGLT2 upon co-expression of MAP17 (Fig. 5e). These data suggest that MAP17 acts as a structural building block of the hSGLT2-MAP17 complex and could somehow promote the surface expression of hSGLT2. In contrast, the basal uptake activity of hSGLT1 was much higher than that of hSGLT2, and the increase in the uptake activity of hSGLT1 upon co-expression of MAP17 was not as dramatic as that of hSGLT2 (Fig. 5e).

## Discussion

It is proposed that SGLT1 has two sodium-binding sites (Na2 and Na3) and SGLT2 has one sodium-binding site (Na2)[16]. The electron density of sodium ion could be observed in the Na2 site of the 3 Å hSGLT2-empagliflozin map (EMD-31558)[10] when further sharpening was applied (Supplementary Fig. 9a). However, the sodium density could not be observed in the map of the hSGLT1-LX2761

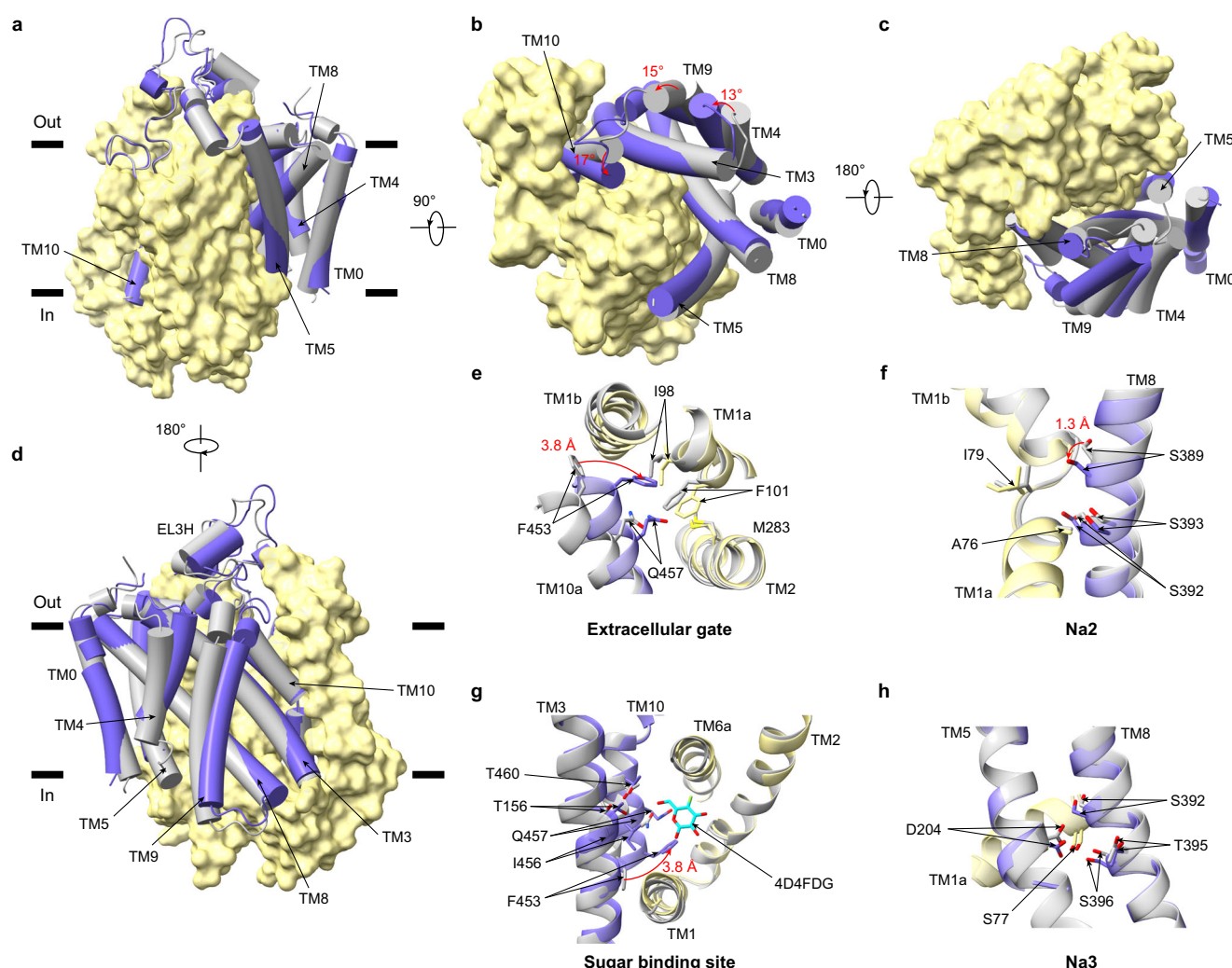

**Fig. 3 | Structural changes from the outward-open to the occluded conformation. a** Superposition of hSGLT1$_{outward-open}$ (grey, PDB ID: 7WMV) and hSGLT1$_{occluded}$ (colored). The less-mobile region is shown as a yellow surface, and the moving region is shown as cartoons. Helices are shown as cylinders. **b** The top view of hSGLT1, the movements of helices are shown as red arrows. **c** The bottom view of hSGLT1. **d** The 180° rotation of (**a**). **e** The movements of residues at the extracellular gate, the color is the same as (**a**). The movements of residues are indicated as red arrows. **f** Structural comparison of the Na2 site, the color is the same as (**a**). **g** Structural comparison at the sugar substrate-binding site, the color is the same as (**a**). **h** Structural comparison of the Na3 site, the color is the same as (**a**).

complex[11] or the maps reported in the current work, even with further sharpening, probably due to the limited local map quality. Nevertheless, the structures of the sodium-binding sites of hSGLT1/2 highly resemble those of LeuT in different states, in which the sodium sites are intact in the outward-facing and the occluded conformations, but are disrupted in the inward-facing conformation. These structural observations correlate well with the essential roles of sodium ions in the transport cycle of SGLT proteins. The formation and destruction of the sodium-binding sites drive the structural changes of the helices of the moving region, leading to the sequential opening and closing of the extracellular gate and intracellular gate, and the concomitant changes in sugar substrate affinity (Fig. 6). It is reported that disruption of the Na3 site in hSGLT1 by the T395A mutation would decrease the affinities of sodium and substrate, but the T395A mutant could still transport substrate[16], suggesting that the Na3 site is important but not essential for the coupling of the sodium gradient to sugar transport. In agreement with this, we found similar structural changes of the Na2 and Na3 sites in hSGLT2, even though the Na3 site of hSGLT2 is not functional, indicating the Na3 site likely enhances the sodium and sugar affinity thermodynamically (Supplementary Fig. 9b, c). Notably,

it is recently reported that another SLC5 family member, the sodium/iodide symporter (NIS, SLC5A5), might use two Na sites that are structurally different from the Na2 or Na3 site of hSGLT[17], suggesting a distinct structural mechanism for iodide transportation.

Previous mechanistic studies on vSGLT based on homology modeling, MD-simulation, and DEER measurement suggest the bundle domain (TM1, TM2, TM6, and TM7) moves relative to the "hash motif" (TM3, TM4, TM8, and TM9) in a rigid body fashion during conformational transition[14]. However, we found that in hSGLT1 or hSGLT2, the conformation of the moving region (TM0, TM3, TM4, TM5, TM8, TM9, and TM10) changes largely, but the structural changes of the less-mobile region (TM1, TM2, TM6, TM7, TM11, TM12, and TM13) are less pronounced (Supplementary Fig. 6). Particularly, the conformations of TM4 and TM9 within the "hash motif" also change dramatically during state transition in our structures (Supplementary Fig. 6), in contrast to the previous model of vSGLT[14]. The structural changes observed in hSGLT1 and hSGLT2 are also different from those observed in LeuT, in which only TM1, TM2, TM6, and TM7 show large movements during the transport cycle[18,19]. However, the structural changes in hSGLT1 and hSGLT2 are similar to those in BetP to some extent. BetP is a Na-coupled symporter, in which TM1a, TM3, TM4, TM5, TM8, TM9, and

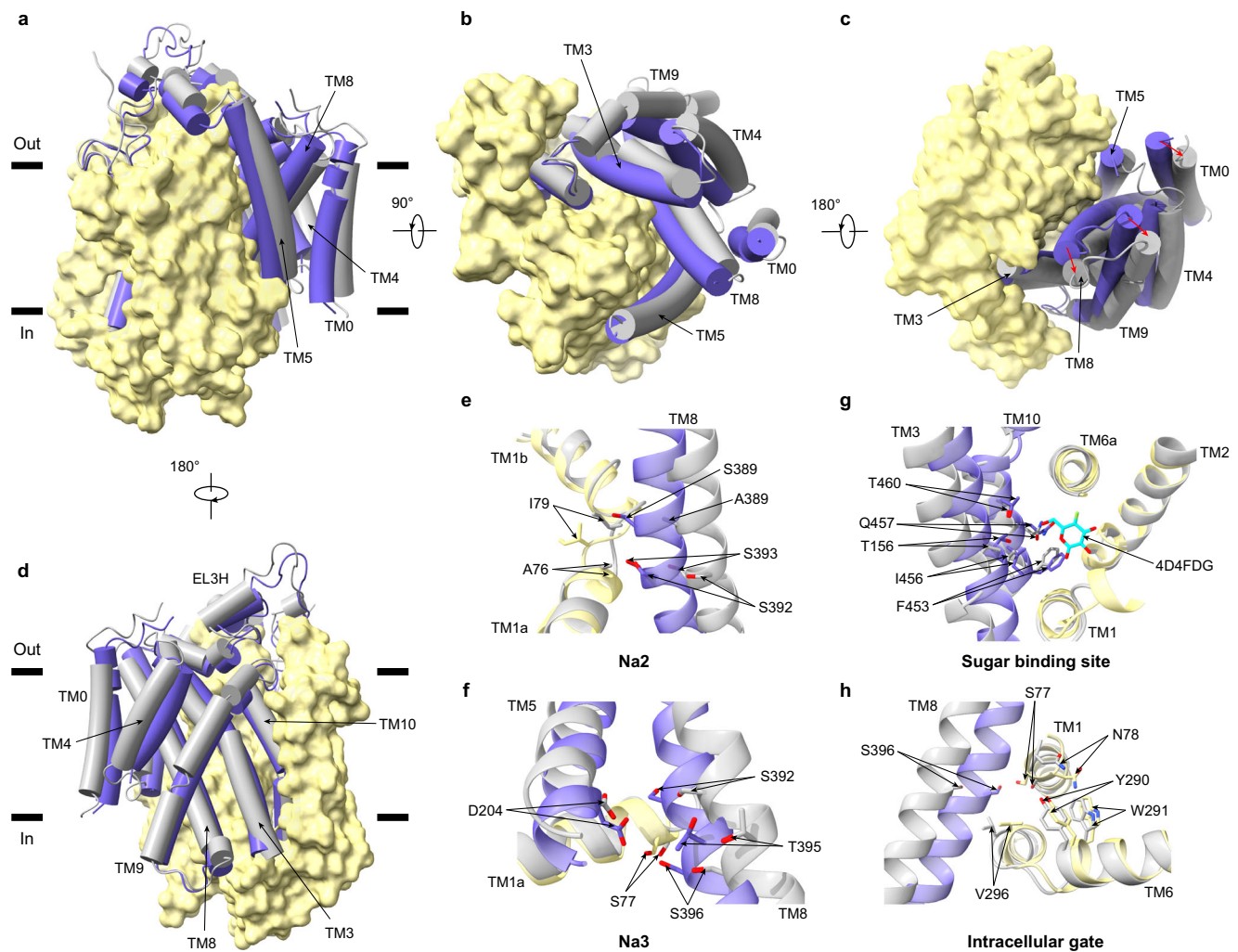

**Fig. 4 | Structural changes from the occluded conformation to the inward-open conformation. a** Superposition of hSGLT1$_{inward-open}$ (grey, PDB ID:7SLA) and hSGLT1$_{occluded}$ (colored). The less-mobile region is shown as a yellow surface, the moving region is shown as cartoons. Helices are shown as cylinders. **b** The top view of hSGLT1, the movements of helices are shown as red arrows. **c** The bottom view of hSGLT1. **d** The 180° rotation of (**a**). **e** The movements of residues at the Na2 site, the color is the same as (**a**). **f** The movements of residues at the Na3 site, the color is the same as (**a**). **g** The movements of residues at the substrate-binding site, the color is the same as (**a**). **h** The movements of residues at the intracellular gate, the color is the same as (**a**).

TM10 show more pronounced movements compared to TM1b, TM2, TM6, and TM7[20]. Our structural observations on hSGLT1/2 further emphasize the diverse structural mechanisms of LeuT-fold sodium-coupled solute transporters.

Although hSGLT2 and hSGLT1 share high structural and sequence similarities, and both of them could interact with MAP17[11], hSGLT2 absolutely requires the auxiliary subunit MAP17 for uptake activity, but hSGLT1 does not[15]. Our structural analysis reveals that MAP17 might play a structural role in the transporter complex because its structure and its interaction with hSGLT stay largely unchanged during the transport cycle. The surface labeling experiment using the antibody against the extracellular side of hSGLT2 showed that robust surface expression of hSGLT2 requires MAP17. This is in contrast to the previous study showing that co-expression of MAP17 did not enhance the surface expression of hSGLT2[15]. However, that study was based on confocal imaging, which might be limited in spatial resolution and less quantitative than our surface labeling experiment. Our structural and functional data favor a model in which hSGLT2 alone might be subjected to intracellular retention and MAP17 is essential for the transportation of hSGLT2 to the plasma membrane to perform the uptake function. In contrast, hSGLT1 alone could traffic to the cell surface to perform sugar uptake.

Taken together, the structures of human SGLTs described here provide a glimpse of the key intermediate occluded state during sugar transportation and provide a solid foundation to further study the mechanisms and dynamics of these transporters.

## Methods

### Cell culture

Sf9 insect cells (Thermo Fisher Scientific, Waltham, MA, USA) were cultured in Sf-900 III serum-free medium (Thermo Fisher Scientific) or SIM SF serum-free medium (Sino Biological) at 27 °C. HEK293F suspension cells (Thermo Fisher Scientific) were cultured in FreeStyle 293 medium (Thermo Fisher Scientific) or SMM 293-TI medium (Sino Biological) supplemented with 1% fetal bovine serum (FBS) at 37 °C with 6% $CO_2$ and 70% humidity. AD293 adherent cells (Agilent Technologies) were cultured in Dulbecco's Modified Eagle Medium (Gibco) supplemented with 10% FBS at 37 °C with 5% $CO_2$. The cell lines were routinely checked to be negative for mycoplasma contamination but have not been authenticated.

### Uptake

1-NBD-glucose uptake assay was used to measure the sugar transport activity of SGLT[10]. AD293 cells cultured in the 12-well

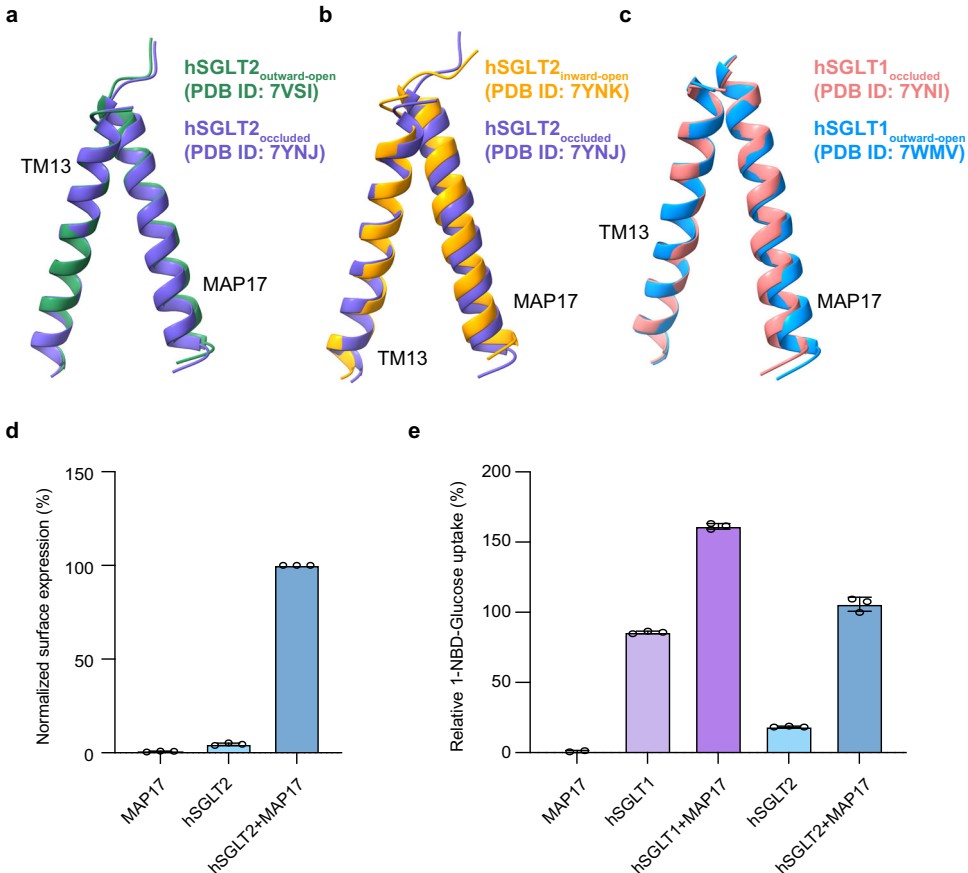

**Fig. 5 | The MAP17 facilitates surface expression of hSGLT2. a** Superposition of TM13 and MAP17 of hSGLT2$_{inward-open}$ (orange) and hSGLT2$_{occluded}$ (purple). **b** Superposition of TM13 and MAP17 of hSGLT2$_{outward-open}$ (green) and hSGLT2$_{occluded}$ (purple). **c** Superposition of TM13 and MAP17 of hSGLT1$_{outward-open}$ (blue) and hSGLT1$_{occluded}$ (pink). **d** Surface expression of wild-type hSGLT2 as determined by antibody mAb90 binding, data are normalized to hSGLT2-MAP17.

(Data are shown as means ± standard deviations; $n = 3$ biologically independent experiments.) **e** 1-NBD-glucose uptake of wild-type hSGLT1 and hSGLT2, data are normalized to hSGLT2-MAP17. (Data are shown as means ± standard deviations; $n = 3$ biologically independent experiments.) Source data are provided as a Source Data file.

plates were transfected with wild-type hSGLT1 and hSGLT2 with or without wild-type MAP17. One day post-transfection, cells were seeded into a 96-well plate coated with poly-D-lysine. After attachment, cells were washed with 200 µl per well of PBS (10 mM Na$_2$HPO$_4$, 2 mM KH$_2$PO$_4$, 137 mM NaCl, and 2.7 mM KCl) twice, followed by incubation at 37 °C with 5% CO$_2$ for 1 h in uptake buffer (10 mM HEPES pH 7.4, 150 mM NaCl, 1 mM CaCl$_2$, and 1 mM MgCl$_2$) supplemented with 600 µM 1-NBD-glucose and 0.3% bovine serum albumin (BSA). Subsequently, cells were washed three times with 200 µl per well of PBS to stop the uptake. Then the cells were lysed with 150 µl per well of TBS buffer containing 1% SDS for 30 min at 25 °C. The lysates were transferred to a clear-bottom black 96-well plate for 1-NBG-glucose detection with excitation at 445 nm, and emission at 525 nm on an Infinite 200Pro imager (Tecan Life Sciences). Protein concentration in 96-well plate was determined using a BCA protein assay kit (CWBIO). The 1-NBD-glucose fluorescence signals in each well were normalized to the protein concentration. The specific 1-NBD-glucose uptake of each well was calculated by subtracting NBD fluorescence signals in wells transfected with empty vector. The IC$_{50}$ of 4D4FDG was calculated by a three parameters Dose-response equation in GraphPad Prism.

### Surface labeling using antibodies
Monoclonal antibody mAb90 was generated by injecting purified hSGLT2-MAP17 protein into mice, followed by hybridoma screening.

The work was carried out by Proteintech Group. The surfacing labeling was carried out as described previously[21]. AD293 cells were plated onto a poly-D-lysine-treated 24-well plate and transfected with wild-type hSGLT2 with or without MAP17 and incubated for 40–48 h. The cells were washed with 500 µl PBS per well twice, fixed using 300 µl 4% formaldehyde in PBS for 30 min, and washed with 500 µl PBS per well twice again. For surface expression, the cells were blocked with 300 µl 3% goat serum in PBS per well for 30 min and labeled with 150 µl primary antibody mAb90 (diluted 3350 times in blocking buffer) per well for 1 h. After washing with PBS three times, cells were incubated with horseradish-peroxidase (HRP) labeled goat anti-mouse IgG secondary antibody (#31444; Invitrogen; the antibody was diluted 2500 times in blocking buffer) for 30 min. After extensive washing, the cells were incubated with High-Sig ECL Western Blotting Substrate (Tanon) for 2 min, and chemiluminescence signals were measured with an Infinite M Plex plate reader (Tecan).

### Protein expression, purification and nanodisc reconstitution
The expression and purification of the hSGLT1$_{GFP}$-MAP17$_{nb}$ complex and the hSGLT2$_{GFP}$-MAP17$_{nb}$ complex were carried out as described previously with minor modifications[10,11]. BacMam virus was generated using sf9 cells. For protein expression, HEK293F cells cultured in SMM 293-TI medium at a density of $2.5 \times 10^6$ cells per ml were infected with 10% of the volume of P2 virus. Sodium butyrate (10 mM) was added to the culture 12 h after infection to promote protein expression, and the cells were transferred to a 30 °C incubator for an additional 36 h before

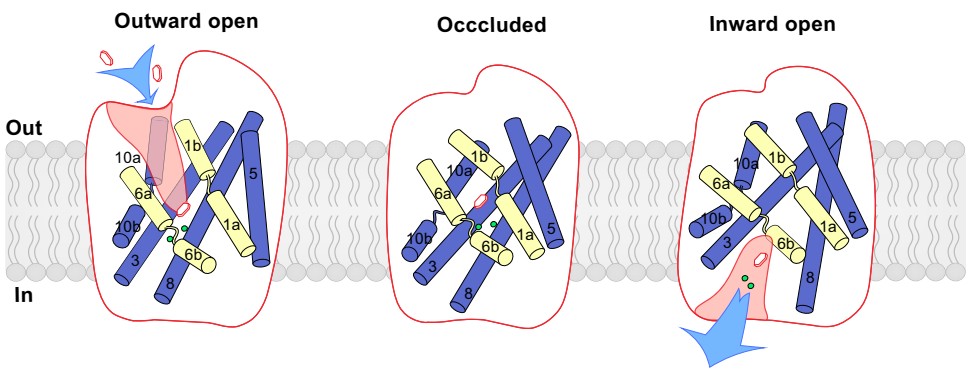

**Fig. 6 | Working model of sugar transport by SGLT.** SGLT1 embedded in the membrane is shown as cartoons. Two gating helices are shown as yellow cylinders.

Four helices in the moving region are shown as blue cylinders. Sugar substrates were shown as hexagons, and sodium ions are shown as green spheres.

harvesting. Cells were collected by centrifugation at 3999×*g* (JLA-8.1000, Beckman Coulter) for 10 min at 4 °C and washed with TBS buffer (20 mM Tris pH 8.0 at 4 °C, 150 mM NaCl) containing 2 μg/ml aprotinin, 2 μg/ml pepstatin and 2 μg/ml leupeptin. The cells were then flash-frozen and stored at −80 °C.

For purification of the hSGLT1$_{GFP}$-MAP17$_{nb}$, the membrane pellets corresponding to 1 liter of culture were resuspended and homogenized in 15 ml TBS buffer containing protease inhibitors (2 μg/ml aprotinin, 2 μg/ml pepstatin, 2 μg/ml leupeptin, and 1 mM phenylmethanesulfonyl fluoride (PMSF)), 10 mM MgCl$_2$, 0.7 μg/ml benzonase, and 5 mM 4D4FDG (Toronto Research Chemicals). The crude membrane solution was incubated at 37 °C for 1 h to facilitate 4D4FDG binding. Digitonin (BioSynth) was added to a final concentration of 1% (W/V) after cooling the membrane to 4 °C, and the mixture was stirred for 1 h to solubilize membrane proteins. The insoluble debris was removed by centrifugation at 193,400 × *g* (Type 50.2 Ti, Beckman Coulter) for 30 min. Subsequently, the supernatant was loaded onto a 5 ml Streptactin Beads 4FF (Smart Lifesciences) column and washed with 50 ml wash buffer 1 (TBS buffer plus 40 μM glyco-diosgenin (GDN, Anatrace), 5 mM 4D4FDG, 10 mM MgCl$_2$ and 1 mM adenosine triphosphate (ATP)) to remove the contamination of heat shock proteins. Then, the column was extensively washed with 50 ml wash buffer 2 (TBS buffer supplemented with 40 μM GDN and 5 mM 4D4FDG). The target protein was eluted with elution buffer containing 50 mM Tris pH 8.0 at 25 °C, 150 mM NaCl, 40 μM GDN, 5 mM 4D4FDG, and 10 mM D-desthiobiotin (IBA). The eluate was loaded onto a HiTrap Q HP column (GE Healthcare) and the hSGLT1-MAP17 complex was separated from aggregates with a linear gradient from 0 mM NaCl to 1000 mM NaCl in buffer containing 20 mM Tris pH 8.0 at 4 °C, 40 μM GDN, and 5 mM 4D4FDG. The fractions containing the hSGLT1-MAP17 complex were collected for nanodisc reconstitution. MSP2X which contains two copies of MSP1E3D1 was purified according to the previously described method[22]. The hSGLT1-MAP17 complex was mixed with MSP2X and soybean polar lipids extract (SPLE, solubilized in 1% (W/V) GDN, Avanti) at a molar ratio of protein: MSP2X: SPLE = 1: 4: 400. The mixture was allowed to equilibrate for 1 h at 4 °C, and then Bio-beads SM2 (Bio-Rad) were added to initiate the reconstitution with constant rotation at 4 °C. Bio-beads SM2 were added to the mixture four times within 24 h to gradually remove detergents from the system. Afterward, the proteins that were not reconstituted in lipid nanodisc were removed by centrifugation at 86,600 × *g* for 30 min in TLA 100.3 rotor (Beckman Coulter). The supernatant containing the hSGLT1-MAP17 complex reconstituted in lipid nanodisc was loaded onto the 1 ml Streptactin Beads 4FF column to remove empty nanodisc. The elution from the Streptactin Beads 4FF column was concentrated and subjected to a Superose 6 increase 10/300 GL column (GE Healthcare) in the buffer that contained 20 mM HEPES pH 7.5, 150 mM NaCl and 5 mM 4D4FDG. The peak fractions corresponding to the hSGLT1-MAP17 complex in

lipid nanodiscs were collected and diluted to a concentration of approximately 0.1 mg/ml for cryo-EM sample preparation.

For purification of the hSGLT2$_{GFP}$-MAP17$_{nb}$ complex, the membrane pellets were resuspended and homogenized in TBS buffer containing protease inhibitors, 10 mM MgCl$_2$ and 0.7 μg/ml benzonase. The crude membrane solution was incubated at 37 °C for 1 h and then cooled to 4 °C. The hSGLT2-MAP17 complex was solubilized by 1% (W/V) GDN for 1 h with constant rotation. The insolubilized materials were removed by centrifugation at 193,400 × *g* for 30 min in Ti50.2 rotor (Beckman Coulter). The supernatant was loaded onto a Streptactin Beads 4FF (Smart Lifesciences) column. The column was washed sequentially with buffer W1 (TBS buffer plus 40 μM GDN, 10 mM MgCl$_2$ and 1 mM ATP) and buffer W2 (TBS buffer plus 40 μM GDN). The hSGLT2-MAP17 complex was eluted with buffer E (50 mM Tris pH 8.0 at 25 °C, 150 mM NaCl, 40 μM GDN and 10 mM D-desthiobiotin (IBA)). The hSGLT2-MAP17 complex in elution was further purified by a HiTrap Q HP (GE Healthcare) column to remove D-desthiobiotin. The fractions from the HiTrap Q HP column, which contained the hSGLT2-MAP17 complex, were collected for nanodisc reconstitution. The hSGLT2-MAP17 complex was mixed with SPLE (solubilized in 1% GDN) and NW9 at a molar ratio of protein: NW9: SPLE = 1: 2: 100. The mixture was incubated for 1 h at 4 °C, followed by adding Bio-beads SM2 (Bio-Rad) to initiate the reconstitution. A total of four batches of Bio-beads SM2 were added during the reconstitution process. After reconstitution, the hSGLT2-MAP17 complex in lipid nanodisc was clarified by centrifugation at 86,600 × *g* for 30 min in TLA 100.3 rotor (Beckman Coulter). The empty nanodisc was removed by a Streptactin Beads 4FF column according to the previously described method in buffer without detergent. The hSGLT2-MAP17 complex reconstituted in lipid nanodisc was further purified by a Superose 6 increase 10/300 GL column (GE Healthcare) in TBS buffer. The peak fractions were collected. For the hSGLT2$_{GFP}$-MAP17$_{nb}$ complex in the apo state, the protein was diluted to a concentration of approximately 0.1 mg/ml for cryo-EM sample preparation. For the hSGLT2$_{GFP}$-MAP17$_{nb}$ complex in the AMG binding state, the protein was incubated with 100 mM AMG for 1 h at 37 °C for substrate binding before cryo-EM sample preparation.

## Cryo-EM sample preparation and data collection
The surface of Quantifoil Au 300 mesh R 0.6/1.0 grids were coated with graphene oxide[23]. Aliquots of 2.5 μl of the hSGLT1-MAP17 complex or the hSGLT2-MAP17 complex in presence of 0.5 mM fluorinated octylmaltoside (FOM, Anatrace) were applied to the grids. After incubation at 4 °C under 100% humidity for 60 s, the grids were blotted for 4 s using a blot force of 4, and then plunge-frozen into liquid ethane using a Vitrobot Mark IV (Thermo Fisher Scientific). The grids were transferred to a Titan Krios electron microscope (Thermo Fisher) operating at 300 kV. SerialEM-3.6.11 was used for automated data collection.

Movies from the dataset of the hSGLT1-MAP17 were recorded on a K3 Summit direct detector (Gatan) mounted post a quantum energy filter (slit width 20 eV) in super-resolution mode with a defocus range of −1.4 to −1.8 μm and a magnification of 105,000×, resulting in a calibrated pixel size of 0.417 Å. Movies from datasets of the hSGLT2-MAP17 in the apo state and the AMG binding state were collected on a K2 Summit direct detector (Gatan) at a magnification of 165,000× with a calibrated pixel size of 0.4105 Å. Each stack of 32 frames was exposed for 8 s, with an exposing time of 0.25 s per frame at a dose rate of 4.7 electrons per Å$^{-2}$ per second.

## Cryo-EM image processing
For the hSGLT1-MAP17 in complex with 4D4FDG, a total of 4600 micrographs were collected. Beam-induced drift was corrected using MotionCor2[24] and 2× binned to a pixel size of 0.834 Å. The contrast transfer function (CTF) parameters of dose-weighted micrographs were estimated by Gctf[25]. Micrographs were manually screened, and a total of 3746 micrographs were used for the following process. A total of 2,794,295 particles were auto-picked using Gautomatch-0.56 (developed by K. Zhang). All particles were extracted with a box size of 120 and 2× binned (pixel size 1.668 Å) in Relion-3.1[26]. These particles were subjected to 2D classification with cryoSPARC-3.1.0[27]. After two rounds of 2D classification, 537,777 particles from classes exhibiting recognizable hSGLT1 features were selected for further processing. A batch size of 50,000 particles was used to generate an initial 3D reference model. The initial model was low-pass filtered to resolutions of 8 Å, 15 Å, 25 Å and 35 Å. The low-pass filtered maps were used as references for a heterogeneous refinement without symmetry imposed. Subsequently, 246,748 particles from the best classes were re-extracted using a box size of 240 at 0.834 Å per pixel and re-centered in Relion-3.1 and then refined with NU-refinement and local refinement subsequently in cryoSPARC-3.1.0[28], resulting in a reconstruction with an overall resolution of 3.15 Å. Further, seed-facilitated 3D classification[29] was performed in cryoSPARC-3.1.0 and a set of 788,321 particles was enriched. NU-refinement and local refinement of these particles yielded a reconstruction at 2.52 Å. This model was low-pass filtered to resolutions of 4 Å, 8 Å, 12 Å, 16 Å and 20 Å. The maps were used as references for a multi-reference 3D classification with an adapted mask. The classes of the last 12 iterations with the best features of hSGLT1 and MAP17 in 3D classification were combined, and the duplicated particles were removed. The final 318,616 particles were re-extracted using a box size of 240 and a pixel size of 0.834 Å in Relion-3.1. These particles were subjected to NU-refinement and local refinement in cryoSPARC-3.1.0, yielding a reconstruction at 3.26 Å. The resolution was estimated using the Fourier shell correlation (FSC) = 0.143 criterion. Local resolution was calculated using cryoSPARC-3.1.0. ThreeDFSC curves were calculated using the 3DFSC server[30] (https://3dfsc.salk.edu/).

For the hSGLT2-MAP17 complex, a total of 9841 and 7393 micrographs of the hSGLT2-MAP17 complex in the apo state and the AMG binding state were collected respectively, and the beam-induced drift was corrected using MotionCor2[24] and binned to a pixel size of 0.821 Å. Dose-weighted micrographs were used for CTF estimation by Gctf[25]. A total of 3,313,984 particles (the apo state) and 3,310,006 particles (the AMG binding state) were auto-picked using Gautomatch-0.56 (developed by K. Zhang). Two rounds of 2D classification in CryoSPARC-3.1.0[27] were performed to remove ice spots, contaminants and aggregates, yielding 139,915 and 301,506 particles for the apo state and the AMG binding state, respectively. After ab-initio reconstruction and heterogeneous refinement using CryoSPARC-3.1.0[27], 25,599 particles (the apo state) and 124,469 particles (the AMG binding state) were selected and used as seeds for seed-facilitated classification. After that, 107,441 particles (the apo state) and 647,871 particles (the AMG binding state) were retained and yielded reconstructions at 3.61 Å and 3.61 Å, respectively. To further improve the map quality of the hSGLT2-MAP17 complex, multi-reference 3D classification without alignment was performed with an adapted mask, resulting in a reconstruction at 3.48 Å from 39,476 particles for the apo state and 3.79 Å from 31,089 particles for the AMG binding state. After that, the particles were re-extracted from micrographs of AMG binding state, and the map of 3.79 Å was input as a seed. In total, 71,638 particles were extracted and performed a 2D classification, leaving 44,393 particles. After NU refinement and local refinement, a map at 3.33 Å was obtained. Resolutions were estimated using the gold-standard Fourier shell correlation criterion of 0.143. Local resolution was calculated using CryoSPARC-3.1.0[27].

## Model building
The homology model of the hSGLT1-MAP17 complex in 4D4FDG binding state was from the hSGLT1-MAP17 complex with LX2761 (PDB ID: 7WMV). The models of the hSGLT2-MAP17 complex in the apo state and the AMG binding state were from the model of the hSGLT2-MAP17 complex with empagliflozin (PDB ID: 7VSI). The models were fitted into the cryo-EM maps using UCSF Chimera[31] and manually adjusted using Coot-0.8.9.3[32]. The substrates were manually fitted into the electron densities. The models were further refined against the maps using Phenix-1.19[33]. The RMSD between different states were calculated with Pymol-1.7.0.5.

## Quantification and statistical analysis
Global resolution estimations of cryo-EM density maps are based on the 0.143 Fourier Shell Correlation criterion[34]. The local resolution was estimated using Relion-3.1[26]. The number of independent reactions (N) and the relevant statistical parameters for each experiment (such as mean or standard deviation) are described in the figure legends. No statistical methods were used to pre-determine sample sizes.

## Reporting summary
Further information on research design is available in the Nature Portfolio Reporting Summary linked to this article.

## Data availability
The data that support the findings of this study are available from the corresponding author upon request. Cryo-EM maps and atomic coordinates of the hSGLT1-MAP17 complex in 4D4FDG binding state have been deposited in the EMDB and PDB database under the ID codes EMDB: EMD-33962 and PDB: 7YNI, respectively. Cryo-EM maps and atomic coordinates of the hSGLT2-MAP17 complex in apo state and AMG binding state have been deposited in the EMDB and PDB database under the ID codes EMDB: EMD-33964, EMD-33963 and PDB: 7YNK, 7YNJ, respectively. EMDB entry (EMD-31558) and PDB entries (7WMV, 3DH4, 7SLA and 7VSI) used in this study were downloaded from EM Data Bank and Protein Data Bank, respectively. Source data are provided with this paper.

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

## Acknowledgements

Cryo-EM data collection was supported by Electron microscopy laboratory and Cryo-EM platform of Peking University with the assistance of Xuemei Li, Changdong Qin, Xia Pei, Zhenxi Guo, Xiaojuan Hui, and Guopeng Wang. Part of the structural computation was also performed on the Computing Platform of the Center for Life Science and High-performance Computing Platform of Peking University. We thank the National Center for Protein Sciences at Peking University in Beijing, China for assistance with negative stain EM. The work is supported by grants from the Ministry of Science and Technology of China (National Key R&D Program of China, 2022YFA0806504 to L.C.), the National Natural Science Foundation of China (91957201, 32225027, and 31821091 to L.C.) and Center For Life Sciences (CLS to L.C.).

## Author contributions

L.C. initiated the project and wrote the manuscript draft. W.C. and Y.N. purified the protein, prepared the cryo-EM sample. Y.N., W.C., and R.L. collected the cryo-EM data. Y.N. and W.C. processed data. Y.N., W.C., and L.C. built and refined the model. W.C. and Z.S. performed uptake assay. All authors contributed to the manuscript preparation.

## Competing interests

The authors declare no competing interests.
