## [Peer Review File · Nature Communications]

Structures of human SGLT in the occluded state reveal conformational changes during sugar transportREVIEWER COMMENTS

Reviewer #1 (Remarks to the Author):

Cui et. al. report SGLTs structures in substrate bound and occlude state, which fill a knowledge gap in our understanding of SGLT transport cycle. This manuscript could be further improved.

1. It may be worthwhile to perform molecular dynamic simulation to confirm the binding pose of substrate. The substrate pose may be ambiguous at this resolution
2. Is there an explanation for the higher affinity of fluorinated glucose than glucose based on the structures?
3. It may be useful to include density maps for the helices of SGLTs.
4. Some structure-based functional studies could be helpful.

Reviewer #2 (Remarks to the Author):

The authors present the Cryo EM structures of SGLT2-MAP17 (sugar AMG) and SGLT1(sugar 4FG) bound to synthetic sugars that are reported to have stabilised an occluded conformation. While the structures represent an important mechanistic achievement, I find myself a wee bit disappointed both by the lack of mechanistic discussion and the technical quality of the work presented. While I am willing to give the authors the benefit of my doubt (perhaps the authors felt rushed due to competition?), the manuscript comes across as premature and has some major issues.

Sugar bound

In the vSGLT structure bound to galactose, it has been shown that all OH-groups are hydrogen-bonded to side-chain residues, which is typically for monosaccharide coordination. A comparison of the vSGLT and SGT1 binding site has lead to selective mutations that shift sugar preference based on the same predicted binding mode (Nature 601:275) (JMB 434 (2022)). As such, the expected binding pose for D.glucose in SGLT1/2 is thought to closely match the D-galactose coordination in vSGLT. The modelled residues to human SGLT1 are thought to be made up of K321, E102, W291, T287, F101, Q457, and H83 (JMB 434 (2022)). However, in Fig. 1c it seems if's unclear in any of these residues are actually within hydrogen bonding distance to the bound sugar? I couldnt find a figure showing the AMG in SGLT2? Please include a structural superimposition of D-galactose in SGLT2 with vSGLT? Also the authors previously published the structure of SGLT2 in complex with the type 2 drug Jaundice (empagliflozin). The drug is made up of a sugar backbone, which biochemical evidence supports that it binds in a an analogous manner to D-glucose. Please include the comparisons to the coordination vSGLT with galactose and the SGLT2 with the sugar-based inhibitor.

Cryo EM maps

I am concerned about the reported quality of the Cryo EM maps. The density doesn't seem like clear alpha helices. It would certainly help to show how the TM helices fit into the Cryo EM maps by showing individual helices in the supplementary. The cause for concern is supported by the PDB the validation reports highlighting that only 60% of the modelled atoms are fitted into the map density. For a typical Cryo EM structure at 3 to 3.5Å resolution I would expect, at least, around 80% of the residues are supported by map density. Please explain?

Conformational transitions

It would be helpful to compare the structural transitions to other LeuT-fold members were structures have outlined the structural transitions and those more detailed by SGLT. How do the conformational transitions alter the current working model derived from vSGLT by Jeff Abramson and Ernie Wright (PNAS 115 (12) E2742-E2751), for example?

We greatly appreciate the time, efforts, and constructive comments of reviewers for improving our manuscript. During this revision, we have improved the resolution of SGLT2-AMG complex and SGLT2-apo to the resolution of 3.33 Å and 3.48 Å, respectively. We have also included supporting functional data. Please see the point-to-point responses below (colored in dark blue).

Reviewer #1 (Remarks to the Author):

Cui et. al. report SGLTs structures in substrate bound and occlude state, which fill a knowledge gap in our understanding of SGLT transport cycle. This manuscript could be further improved.

1. It may be worthwhile to perform molecular dynamic simulation to confirm the binding pose of substrate. The substrate pose may be ambiguous at this resolution.

Response: We agree with the reviewers' point: at this resolution, the modeling of small sugar substrates would be difficult purely based on electron density maps. In this work, we modeled the sugar substrates based on the binding pose of the X-ray structure of vSGLT in complex with galactose (PDB ID: 3DH4) and SGLT2 in complex with empagliflozin (PDB ID: 7VSI). The detailed modeling process was included in the revised manuscript from line 81 to line 84. Since the exact binding poses of sugar substrates could not be explicitly determined from the experimental map, we intentionally not focused our discussion on this topic to avoid over-interpretation. We have also warned readers on this point from line 87 to line 93: "Notably, although the structural models, especially the coordinates of sugar substrates, were refined against the cryo-EM maps to reasonable geometry, we suggest a cautious interpretation of the sugar-binding pose, because of the small size of sugars, the indistinguishable density of their hydroxyl groups, and the large positional uncertainty intrinsic to this resolution range (~3.3Å)". We think the exact binding pose of the sugar substrate would be experimentally determined from structures with higher resolutions in further studies.

2. Is there an explanation for the higher affinity of fluorinated glucose than glucose based on the structures?

Response: Based on our structural model, the fluorine atom of 4D4FDG might form a hydrogen bond with T287 of SGLT1. It is possible that this F-T287 hydrogen bond might be more stable than the hydroxyl-T287 bond when glucose is bound, explaining why 4D4FDG has lower a K_m . But we don't have data supporting this hypothesis and did not incorporate this speculation in the revised manuscript.

3. It may be useful to include density maps for the helices of SGLTs.

Response: We have provided the density maps for each helix in the revised Supplementary Fig 3, 5 and 8.

4. Some structure-based functional studies could be helpful.

Response: We have provided the 1-NBD-glucose uptake assay using 4D4FDG as the competitor in the revised Figure 1a, validating the binding between 4D4FDG and the cryo-EM construct of hSGLT1. We have also extensively tried to design disulfide bonds to lock SGLT1 in a specific conformation. However, we found the function of wt hSGLT1 itself could be severely affected by redox status, making the interpretation of results ambiguous. Therefore, we did not include these data in the revised manuscript.

Reviewer #2 (Remarks to the Author):

The authors present the Cryo EM structures of SGLT2-MAP17 (sugar AMG) and SGLT1(sugar 4FG) bound to synthetic sugars that are reported to have stabilised an occluded conformation. While the structures represent an important mechanistic achievement, I find myself a wee bit disappointed both by the lack of mechanistic discussion and the technical quality of the work presented. While I am willing to give the authors the benefit of my doubt (perhaps the authors felt rushed due to competition?), the manuscript comes across as premature and has some major issues.

Sugar bound

In the vSGLT structure bound to galactose, it has been shown that all OH-groups are hydrogen-bonded to side-chain residues, which is typically for monosaccharide coordination. A comparison of the vSGLT and SGT1 binding site has lead to selective mutations that shift sugar preference based on the same predicted binding mode (Nature 601:275) (JMB 434 (2022)). As such, the expected binding pose for D.glucose in SGLT1/2 is thought to closely match the D-galactose coordination in vSGLT. The modelled residues to human SGLT1 are thought to be made up of K321, E102, W291, T287, F101, Q457, and H83 (JMB 434 (2022)). However, in Fig. 1c it seems ifs unclear in any of these residues are actually within hydrogen bonding distance to the bound sugar?

Response: We modeled the sugar substrates based on the binding pose of the X-ray structure of vSGLT in complex with galactose (PDB ID: 3DH4) and SGLT2 in complex with empagliflozin (PDB ID: 7VSI) as indicated in the revised manuscript from line 81 to line 84. Therefore, the binding modes of these sugars are close to that of the vSGLT-galactose structure. Because the exact binding pose of the sugar substrate could not be explicitly determined from the experimental map, we intentionally not focused our discussion on this topic to avoid over-interpretation. We have also warned readers on this point from line 89 to line 92: “Notably, although the structural models, especially the coordinates of sugar substrates, were refined against the cryo-EM maps to reasonable geometry, we suggest a cautious interpretation of the sugar-binding pose, because of the small size of sugars, the indistinguishable density of their

hydroxyl groups, and the large positional uncertainty intrinsic to this resolution range (~3.3Å)”. We think the exact binding pose of the sugar substrate would be experimentally determined from structures with higher resolutions in further studies.

I couldnt find a figure showing the AMG in SGLT2?

Response: The density of AMG bound in SGLT2 is shown in the revised Supplementary Fig. 5.

Please include a structural superimposition of D-galactose in SGLT2 with vSGLT?

Also the authors previously published the structure of SGLT2 in complex with the type 2 drug Jaundice (empagliflozin). The drug is made up of a sugar backbone, which biochemical evidence supports that it binds in a an analogous manner to D-glucose. Please include the comparisons to the coordination vSGLT with galactose and the SGLT2 with the sugar-based inhibitor.

Response: We have included the structure comparison of 4D4FDG with galactose in vSGLT in the revised Supplementary Fig. 3. We have also included the structure comparison of AMG with empagliflozin in SGLT2 in the revised Supplementary Fig. 5.

Cryo EM maps

I am concerned about the reported quality of the Cryo EM maps. The density doesn't seem like clear alpha helices. It would certainly help to show how the TM helices fit into the Cryo EM maps by showing individual helices in the supplementary. The cause for concern is supported by the PDB the validation reports highlighting that only 60% of the modelled atoms are fitted into the map density. For a typical Cryo EM structure at 3 to 3.5Å resolution I would expect, at least, around 80% of the residues are supported by map density. Please explain?

Response: The contour level in the originally submitted map was incorrectly set. Please check the updated PDB validation report uploaded to the system.

Conformational transitions

It would be helpful to compare the structural transitions to other LeuT-fold members were structures have outlined the structural transitions and those more detailed by SGLT. How do the conformational transitions alter the current working model derived from vSGLT by Jeff Abramson and Ernie Wright (PNAS 115 (12) E2742-E2751), for example?

In regards to the enquiry the simple answer is both. Firstly, how do these structural transitions fit in with the models predicted by vSGLT structures?

Response: The pioneering work by Jeff Abramson and Ernest Wright in PNAS 115 (12) E2742-E2751 was based on homology modeling and MD simulations and had several drawbacks:

1. That work used the structure of SaiT (PDB ID: 5NV9) for the homology modelling of the outward-open state. However, SaiT (5NV9) has large a structural difference compared with SGLT1 (7MWV) or SGLT2 (7VSI) in the outward-open state, questioning the suitability of using SaiT as the homology model.

2, The occluded state structure of vSGLT used in that work (3DH4) is the inward-facing occluded state which is similar to the inward-open state regarding the overall conformation (2XQ2). Moreover, that structure (3DH4) is very different from the occluded state structure determined in our current work, which is occluded to both sides. Therefore, previous work did not capture the conformational changes from the outward-facing to the occluded state or from the occluded state to the inward-open state.

In addition, our work differs from previous work in the details of conformational change during state transition. Particularly, previous work suggests a model where the “hash motif” (TM3, TM4, TM8, and TM9) moves as a rigid body during the transport cycle but we observed that there is a large structural difference within the hash motif and TM4 and TM9 move in relative to TM3 and TM8.

Secondly, on a more general level how do these transitions differ from other LeuT members?

Response: we have included the comparison with BetP, a Na-coupled symporter in the revised manuscript from line 198 to line 202. In BetP, M1a, M3, M4, M5, M8, M9, and M10 show more pronounced movements compared to M1b, M2, M6, and M7. Except for M1, other TM helices (TM2-TM10) of BetP show similar trends of movements compared to SGLT1/2.

Clearly it will be important for the authors to discuss the sodium sites, which differ between LeuT members and drive different local changes. For SGLT1 and 2 it is the non-conserved Na3 site, which is of more interest in the field and how this might enable the formation of the occluded state.

Response: In SGLT2, there is only one sodium site (Na2). In SGLT1, there are two sites (Na2 and Na3). Previous results have shown that mutation of Na3 of SGLT1 would not abolish the sugar transporting function but will decrease the sodium and sugar affinity of SGLT1 (Bisignano et al., Nat Commun, 2018, PMID: 30532032). In our structures, we found similar structural changes of Na2 and Na3 sites in both SGLT1 and SGLT2 as shown in the revised Fig.3-4 and Supplementary Fig. 9. Therefore, we speculate that the Na3 site thermodynamically enhances the sodium and sugar affinity of SGLT1. We have included these discussions from line 178 to line 187.

References:

Bisignano, P., Ghezzi, C., Jo, H., Polizzi, N.F., Althoff, T., Kalyanaraman, C., Friemann, R., Jacobson, M.P., Wright, E.M., and Grabe, M. (2018). Inhibitor binding mode and allosteric regulation of Na(+)-glucose symporters. *Nat Commun* 9, 5245.

REVIEWERS' COMMENTS

Reviewer #1 (Remarks to the Author):

Thanks the authors for addressing our my comments. I have no more concerns.

Reviewer #2 (Remarks to the Author):

Thank you for the extensive revision. I agree that the cryo EM maps are of sufficient quality to conclude they are sugar-bound states, but not good enough to assign their exact coordination. However, by focusing on conformational changes (similarities and differences to previous models), I think there is enough impact beyond substrate coordination per se.

Overall, I think all my previous concerns have been answered.

Optional point for discussion: SGLT1 is known to be a passive transporter for water and a model was proposed in the previous SGLT1 structure (Nature volume 601, pages 274–279 (2022)). We now have an occluded SGLT1 state and there doesn't seem to be any obvious water pathway by the surface representations shown here. Im wondering how much space is around the sugar in the occluded state? Could it be so simple that the sugar drags water into this pocket, which may explain the lack of close coordination by side-chain residues?

We greatly appreciate the time, efforts, and constructive comments of reviewers for improving our manuscript. Please see the point-to-point responses below (colored in dark blue).

Reviewer #1 (Remarks to the Author):

Thanks the authors for addressing our my comments. I have no more concerns.

Reviewer #2 (Remarks to the Author):

Thank you for the extensive revision. I agree that the cryo EM maps are of sufficient quality to conclude they are sugar-bound states, but not good enough to assign their exact coordination. However, by focusing on conformational changes (similarities and differences to previous models), I think there is enough impact beyond substrate coordination per se.

Overall, I think all my previous concerns have been answered.

Optional point for discussion: SGLT1 is known to be a passive transporter for water and a model was proposed in the previous SGLT1 structure (Nature volume 601, pages 274–279 (2022)). We now have an occluded SGLT1 state and there doesn't seem to be any obvious water pathway by the surface representations shown here. Im wondering how much space is around the sugar in the occluded state? Could it be so simple that the sugar drags water into this pocket, which may explain the lack of close coordination by side-chain residues?

Response: We did not observe the continuous water pathway in the occluded state structure of SGLT1. It is possible that the water pathway only opens in certain states, such as the inward-open state as proposed in Nature volume 601, pages 274–279 (2022).

Due to the resolution limitation, we could not observe the water molecules in the cryo-EM maps. It is possible that some water molecules are transported along with sugar substrates, but we do not have any evidence supporting this hypothesis at the moment.